# Information-Theoretic Generalization Bounds for Iterative Semi-Supervised Learning

## ABSTRACT

We consider iterative semi-supervised learning (SSL) algorithms that iteratively generate pseudo-labels for a large amount unlabelled data to progressively refine the model parameters. In particular, we seek to understand the behaviour of the *generalization error* of iterative SSL algorithms using information-theoretic principles. To obtain bounds that are amenable to numerical evaluation, we first work with a simple model—namely, the binary Gaussian mixture model. Our theoretical results suggest that when the class conditional variances are not too large, the upper bound on the generalization error decreases monotonically with the number of iterations, but quickly saturates. The theoretical results on the simple model are corroborated by extensive experiments on several benchmark datasets such as the MNIST and CIFAR datasets in which we notice that the generalization error improves after several pseudo-labelling iterations, but saturates afterwards.

## 1 INTRODUCTION

In real-life machine learning applications, it is relatively easy and cheap to obtain large amounts of unlabelled data, while the number of labelled data examples is usually small due to the high cost of annotating them with true labels. In light of this, semi-supervised learning (SSL) has come to the fore (Chapelle et al., 2006; Zhu, 2008; Van Engelen & Hoos, 2020). SSL makes use of the abundant unlabelled data to augment the performance of learning tasks with few labelled data examples. This has been shown to outperform supervised and unsupervised learning under certain conditions. For example, in a classification problem, the correlation between the additional unlabelled data and the labelled data may help to enhance the accuracy of classifiers. Among the plethora of SSL methods, pseudo-labelling (Lee et al., 2013) has been observed to be a simple and efficient way to improve the generalization performance empirically. In this paper, we consider the problem of pseudo-labelling a subset of the unlabelled data at each iteration based on the previous output parameter and then refining the model progressively, but we are interested in analysing this procedure theoretically. Our goal in this paper is to understand the impact of pseudo-labelling on the generalization error.

A learning algorithm can be viewed as a randomized map from the training dataset to the output model parameter. The output is highly data-dependent and may suffer from overfitting to the given dataset. In statistical learning theory, the *generalization error* is defined as the expected gap between the test and training losses, and is used to measure the extent to which the algorithms overfit to the training data. In SSL problems, the unlabelled data are expected to improve the generalization performance in a certain manner and thus, it is worthwhile to investigate the behaviour theoretically. In this paper, we leverage results in Bu et al. (2020); Wu et al. (2020) to derive an information-theoretic generalization error bound at each iteration for iterative SSL.

We state our main theoretical contribution informally as follows.

**Theorem [Informal]** *For a $d$-variate binary Gaussian mixture model (bGMM) in which each component has variance $\sigma^2$, the generalization error across the different semi-supervised training iterations $|\mathrm{gen}_t|$ can be bounded with high probability as follows:*

$$|\mathrm{gen}_t| \lessapprox \mathrm{const} \cdot \mathbb{E}\left[\sqrt{G_\sigma(F_\sigma^{(t-1)}(\alpha))}\right], \qquad (1)$$

**Figure 1:** Upper bound on $|\mathrm{gen}_t|$ as a function of $t$.

*where $\alpha$ represents the correlation between the optimal and estimated parameter vectors, $F_\sigma^{(t)}$ is the iterated composition of the function $F_\sigma$ (sketched in Figure 3), and $G_\sigma$ (sketched in Figure 5) represents the KL-divergence between the pseudo-labelled and true data distributions.*

As shown in Figure 1, the upper bound is monotonically decreasing in the iteration count $t$ and converges at around $t = 2$ with a sufficiently large amount of unlabelled data. In Section 4, we also show that when the number of labelled data or the variance is large enough, using the unlabelled data does not help to significantly reduce the generalization error across iterations $t$. The behaviour of the empirical generalization error for the bGMM coincides with the upper bound. The results suggest that the proposed upper bound serves as a useful guide to understand how the generalization error changes across the semi-supervised training iterations and it can be used to establish conditions under which unlabelled data can help in terms of generalization. Experimental results on the MNIST and CIFAR datasets corroborate the phenomena for the bGMM that with few labelled data and abundant unlabelled data, the generalization error decreases quickly in the early pseudo-labelling iterations and saturates thereafter. For a more extensive literature review, please refer to Appendix A.

## 2 PROBLEM SETUP

Let the instance space be $\mathcal{Z} = \mathcal{X} \times \mathcal{Y} \subset \mathbb{R}^{d+1}$, the model parameter space be $\Theta$ and the loss fucntion be $l : \mathcal{Z} \times \Theta \to \mathbb{R}$, where $d \in \mathbb{N}$. We are given a labelled training dataset $S_l = \{Z_1, \ldots, Z_n\} = \{(X_i, Y_i)\}_{i=1}^n$ drawn from $\mathcal{Z}$, where each $Z_i = (X_i, Y_i)$ is independently and identically distributed (i.i.d.) from $P_Z = P_{X,Y} \in \mathcal{P}(\mathcal{Z})$ and $X_i$ is i.i.d. from $P_X \in \mathcal{P}(\mathcal{X})$. For any $i \in [n]$, $X_i$ is a vector of features and $Y_i$ is a label indicating the class to which $X_i$ belongs. However, in many real-life machine learning applications, we only have a limited number of labelled data while we have access to a large amount of unlabelled data, which are expensive to annotate. Then we can incorporate the unlabelled training data together with the labelled data to improve the performance of the model. This procedure is called *semi-supervised learning (SSL)*. We are given an independent unlabelled training dataset $S_u = \{X_1', \ldots, X_{\tau m}'\}, \tau \in \mathbb{N}$, where each $X_i'$ is i.i.d. generated from $P_X \in \mathcal{P}(\mathcal{X})$. Typically, $m \gg n$.

In the following, we consider the *iterative self-training with pseudo-labelling* in SSL setup, as shown in Figure 2. Let $t \in [0 : \tau]$ denote the iteration counter. In the initial round ($t = 0$), the labelled data $S_l$ are first used to learn an initial model parameter $\theta_0 \in \Theta$. Next, we split the unlabelled dataset $S_u$ into $\tau$ disjoint equal-size sub-datasets $\{S_{u,k}\}_{k=1}^\tau$, where $S_{u,k} = \{X_{(k-1)m+1}', \ldots, X_{km}'\}$. In each subsequent round $t \in [1 : \tau]$, based on $\theta_{t-1}$ trained from the previous round, we use a predictor $f_{\theta_{t-1}} : \mathcal{X} \mapsto \mathcal{Y}$ to assign a *pseudo-label* $\hat{Y}_i'$ to the unlabelled sample $X_i'$ for all $i \in [(t-1)m + 1 : tm] := \{(t-1)m, (t-1)m+1, \ldots, tm\}$. Let $\hat{S}_{u,t} = \{(X_i', \hat{Y}_i')\}_{i=(t-1)m+1}^{tm}$ denote the $t^{\text{th}}$ pseudo-labelled dataset. After pseudo-labelling, both the labelled data $S_l$ and the pseudo-labelled data $\hat{S}_{u,t}$ are used to learn a new model parameter $\theta_t$. The procedure is then repeated iteratively until the maximum number of iterations $\tau$ is reached.

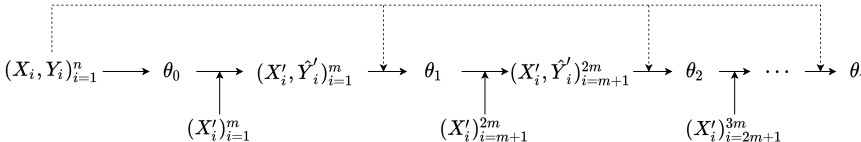

**Figure 2:** Paradigm of iterative self-training with pseudo-labelling in SSL.

Under the setup of iterative SSL, during each iteration $t$, our *goal* is to find a model parameter $\theta_t \in \Theta$ that minimizes the population risk with respect to the underlying data distribution

$$L_{P_Z}(\theta_t) := \mathbb{E}_{Z \sim P_Z}[l(\theta_t, Z)]. \tag{2}$$

Since $P_Z$ is unknown, $L_{P_Z}(\theta_t)$ cannot be computed directly. Hence, we instead minimize the empirical risk. The procedure is termed *empirical risk minimization* (ERM). For any model parameter $\theta_t \in \Theta$, the empirical risk of the labelled data is defined as

$$L_{S_l}(\theta_t) := \frac{1}{n} \sum_{i=1}^n l(\theta_t, Z_i), \tag{3}$$

and for $t \geq 1$, the empirical risk of pseudo-labelled data $\hat{S}_{u,t}$ as

$$L_{\hat{S}_{u,t}}(\theta_t) := \frac{1}{m} \sum_{i=(t-1)m+1}^{tm} l(\theta_t, (X_i', \hat{Y}_i')). \qquad (4)$$

We set $L_{\hat{S}_{u,t}}(\theta_t) = 0$ for $t = 0$. For a fixed weight $w \in [0,1]$, the total empirical risk can be defined as the following linear combination of $L_{S_1}(\theta_t)$ and $L_{\hat{S}_{u,t}}(\theta_t)$:

$$L_{S_1, \hat{S}_{u,t}}(\theta_t) := w L_{S_1}(\theta_t) + (1-w) L_{\hat{S}_{u,t}}(\theta_t). \qquad (5)$$

An SSL algorithm can be characterized by a randomized map from the labelled and unlabelled training data $S_l$, $S_u$ to a model parameter $\theta$ according to a conditional distribution $P_{\theta|S_1,S_u}$. Then at each iteration $t$, we can use the sequence of conditional distributions $\{P_{\theta_k|S_1,S_u}\}_{k=0}^t$ with $P_{\theta_0|S_1,S_u} = P_{\theta_0|S_1}$ to represent an iterative SSL algorithm. The *generalization error* at the $t$-th iteration is defined as the expected gap between the population risk of $\theta_t$ and the empirical risk on the training data:

$$\text{gen}_t(P_Z, P_X, \{P_{\theta_k|S_1,S_u}\}_{k=0}^t, \{f_{\theta_k}\}_{k=0}^{t-1}) := \mathbb{E}[L_{P_Z}(\theta_t) - L_{S_1, \hat{S}_{u,t}}(\theta_t)] \qquad (6)$$

$$= w\left( \mathbb{E}_{\theta_t}[\mathbb{E}_Z[l(\theta_t, Z) \mid \theta_t]] - \frac{1}{n} \sum_{i=1}^n \mathbb{E}_{\theta_t, Z_i}[l(\theta_t, Z_i)] \right)$$

$$+ (1-w)\left( \mathbb{E}_{\theta_t}[\mathbb{E}_Z[l(\theta_t, Z) \mid \theta_t]] - \frac{1}{m} \sum_{i=(t-1)m+1}^{tm} \mathbb{E}_{\theta_t, X_i', \hat{Y}_i'}[l(\theta_t, (X_i', \hat{Y}_i'))] \right). \qquad (7)$$

When $t = 0$ and $w = 1$, the definition of the generalization error above reduces to that of vanilla supervised learning. The generalization error $\text{gen}_t$ is used to measure the extent to which the iterative learning algorithm overfits the training data at the $t$-th iteration. Instead of focusing on the total generalization error induced during the entire process, we are more interested in the following questions. How does $\text{gen}_t$ evolve as the iteration count $t$ increases? Do the unlabelled data examples in $S_u$ help to improve the generalization error?

## 3 PRELIMINARIES

Inspired by the information-theoretic generalization results in Bu et al. (2020, Theorem 1) and Wu et al. (2020, Theorem 1), we derive an upper bound on the generalization error $\text{gen}_t$ for any $t \in [0:\tau]$ in terms of the mutual information between input data samples (either labelled or pseudo-labelled) and the output model parameter $\theta_t$, as well as the KL-divergence between the underlying data distributions and the joint distribution of feature vectors and pseudo-labels.

We denote an $R$-sub-Gaussian random variable $L \in \mathbb{R}$ (Vershynin, 2018) as $L \sim \text{subG}(R)$. Furthermore, let us recall the following non-standard information quantities.

**Definition 1.** *For arbitrary random variables $X$, $Y$ and $U$, define the* disintegrated mutual information *(Negrea et al., 2019; Haghifam et al., 2020) between $X$ and $Y$ given $U$ as $I_U(X;Y) := D(P_{X,Y|U} \| P_{X|U} \otimes P_{Y|U})$, and the* disintegrated KL-divergence *between $P_X$ and $P_Y$ given $U$ as $D_U(P_X \| P_Y) := D(P_{X|U} \| P_{Y|U})$. These are $\sigma(U)$-measurable random variables. It follows immediately that the conditional mutual information $I(X;Y|U) = \mathbb{E}_U[I_U(X;Y)]$ and the conditional KL-divergence $D(P_{X|U} \| P_{Y|U} | P_U) = \mathbb{E}_U[D_U(P_X \| P_Y)]$.*

Let $\theta^{(t)} = (\theta_0, \ldots, \theta_t)$ for any $t \in [0:\tau]$. In iterative SSL, we can upper bound the generalization error as shown in Theorem 1 to follow by applying the law of total expectation.

**Theorem 1** (Generalization error upper bound for iterative SSL). *Suppose $l(\theta, Z) \sim \text{subG}(R)$ under $Z \sim P_Z$ for all $\theta \in \Theta$, then for any $t \in [0:\tau]$,*

$$\left| \text{gen}_t(P_Z, P_X, \{P_{\theta_k|S_1,S_u}\}_{k=0}^t, \{f_{\theta_k}\}_{k=0}^{t-1}) \right| \leq \frac{w}{n} \sum_{i=1}^n \mathbb{E}_{\theta^{(t-1)}}\left[ \sqrt{2R^2 I_{\theta^{(t-1)}}(\theta_t; Z_i)} \right]$$

$$+ \frac{1-w}{m} \sum_{i=(t-1)m+1}^{tm} \mathbb{E}_{\theta^{(t-1)}}\left[ \sqrt{2R^2 \left( I_{\theta^{(t-1)}}(\theta_t; X_i', \hat{Y}_i') + D_{\theta^{(t-1)}}(P_{X_i', \hat{Y}_i'} \| P_Z) \right)} \right]. \qquad (8)$$

The proof of Theorem 1 is provided in Appendix B, in which we provide a general version of upper bound not only applicable for sub-Gaussian loss functions. Compared to Bu et al. (2020, Theorem 1) and Wu et al. (2020, Theorem 1), this bound focuses on the generalization error at each iteration during the learning process, which depends on the disintegrated mutual information and the disintegrated KL-divergence conditioned on the previous outputs. Intuitively, in the upper bound in Theorem 1, the mutual information between the individual input data sample $Z_i$ and the output model parameter $\theta_t$ measures the extent to which the algorithm is sensitive to the input data, and the KL-divergence between the underlying $P_Z$ and pseudo-labelled distribution $P_{X'_i, \hat{Y}'_i}$ measures how well the algorithm generalizes to the true data distribution. As $n \to \infty$ and $m \to \infty$, we show that the disintegrated mutual information $I_{\theta^{(t-1)}}(\theta_t; X'_i, \hat{Y}'_i)$ tends to 0 (in probability), which means that there are sufficient training data such that the algorithm can generalize well. On the other hand, the impact on the generalization error of pseudo-labelling is reflected in the KL-divergence $D_{\theta^{(t-1)}}(P_{X'_i, \hat{Y}'_i} \| P_Z)$ and this term does not necessarily vanish as $n, m \to \infty$. We quantify this precisely in Remark 1 in Section 4.

In iterative learning algorithms, it is usually difficult to directly calculate the mutual information and KL-divergence between the input and the final output (Paninski, 2003; Nguyen et al., 2010; McAllester & Stratos, 2020). However, by applying the law of total expectation and conditioning the information-theoretic quantities on the output model parameters $\theta^{(t-1)} = \{\theta_1, \ldots, \theta_{t-1}\}$ from previous iterations, we are able to calculate the upper bound iteratively. In the next section, we apply the iterated generalization error bound to a classification problem under a specific generative model—the bGMM. This simple model allows us to derive a tractable upper bound on the generalization error as a function of iteration number $t$ that we can compute numerically.

## 4 MAIN RESULTS

We now particularize the iterative semi-supervised classification setup to the bGMM. We calculate the term in (8) to understand the effect of multiple self-training rounds on the generalization error.

Fix a unit vector $\boldsymbol{\mu} \in \mathbb{R}^d$ and a scalar $\sigma \in \mathbb{R}_+ = (0, \infty)$. Under the bGMM with mean $\boldsymbol{\mu}$ and standard deviation $\sigma$ (bGMM$(\boldsymbol{\mu}, \sigma)$), we assume that the distribution of any labelled data example $(\mathbf{X}, Y)$ is specified as follows. Let $\mathcal{Y} = \{-1, +1\}$, $Y \sim P_Y$, where $P_Y(-1) = P_Y(1) = \frac{1}{2}$, and $\mathbf{X}|Y \sim \mathcal{N}(Y\boldsymbol{\mu}, \sigma^2 \mathbf{I}_d)$, where $\mathbf{I}_d$ is the identity matrix of size $d \times d$. In anticipation of leveraging Theorem 1 together with the sub-Gaussianity of the loss function for the bGMM to derive generalization bounds in terms of information-theoretic quantities (just as in Russo & Zou (2016); Xu & Raginsky (2017); Bu et al. (2020)), we find it convenient to show that $\mathbf{X}$ and $l(\boldsymbol{\theta}, (\mathbf{X}, Y))$ are bounded w.h.p.. By defining the $\ell_\infty$ ball $\mathcal{B}^y_r := \{\mathbf{x} \in \mathbb{R}^d : \|\mathbf{x} - y\boldsymbol{\mu}\|_\infty \leq r\}$, we see that

$$\Pr(\mathbf{X} \in \mathcal{B}^Y_r) = \left(1 - 2\Phi\left(-\frac{r}{\sigma}\right)\right)^d =: 1 - \delta_{r,d}, \tag{9}$$

where $\Phi(\cdot)$ is the Gaussian cumulative distribution function. By choosing $r$ appropriately, the failure probability $\delta_{r,d}$ can be made arbitrarily small.

The random vector $\mathbf{X}$ is distributed according to the mixture distribution $p_{\boldsymbol{\mu}} = \frac{1}{2}\mathcal{N}(\boldsymbol{\mu}, \sigma^2 \mathbf{I}_d) + \frac{1}{2}\mathcal{N}(-\boldsymbol{\mu}, \sigma^2 \mathbf{I}_d)$. In the unlabelled dataset $S_\mathrm{u}$, each $\mathbf{X}'_i$ for $i \in [1 : \tau m]$ is drawn i.i.d. from $p_{\boldsymbol{\mu}}$.

For any $\boldsymbol{\theta} \in \Theta$, under the bGMM$(\boldsymbol{\theta}, \sigma)$, the joint distribution of any pair of $(\mathbf{X}, Y) \in \mathcal{Z}$ is given by $\mathcal{N}(Y\boldsymbol{\theta}, \sigma^2 \mathbf{I}_d) \otimes P_Y$. Let the loss function be the *negative log-likelihood*, which can be expressed as

$$l(\boldsymbol{\theta}, (\mathbf{x}, y)) = -\log\left(P_Y(y)p_{\boldsymbol{\theta}}(\mathbf{x}|y)\right) = -\log \frac{1}{2\sqrt{(2\pi)^d}\sigma^d} + \frac{1}{2\sigma^2}(\mathbf{x} - y\boldsymbol{\theta})^\top(\mathbf{x} - y\boldsymbol{\theta}). \tag{10}$$

The minimizer of $\min_{\boldsymbol{\theta} \in \Theta} \mathbb{E}_{(\mathbf{X}, Y) \sim \mathcal{N}(Y\boldsymbol{\mu}, \sigma^2 \mathbf{I}_d) \otimes P_Y}[l(\boldsymbol{\theta}, (\mathbf{X}, Y))]$ is equal to $\boldsymbol{\mu}$. To show that $\boldsymbol{\theta}$ is bounded with high probability, define the set $\Theta_{\boldsymbol{\mu}, c} := \{\boldsymbol{\theta} \in \Theta : \|\boldsymbol{\theta} - \boldsymbol{\mu}\|_\infty \leq c\}$ for some $c > 0$. For any $\boldsymbol{\theta} \in \Theta_{\boldsymbol{\mu}, c}$, we have

$$\min_{(\mathbf{x}, y) \in \mathcal{Z}} l(\boldsymbol{\theta}, (\mathbf{x}, y)) = -\log \frac{1}{2\sqrt{(2\pi)^d}\sigma^d} =: c_1, \quad \text{and} \tag{11}$$

$$\max_{\mathbf{x} \in \mathcal{B}^y_r, y \in \mathcal{Y}} l(\boldsymbol{\theta}, (\mathbf{x}, y)) \leq -\log \frac{1}{2\sqrt{(2\pi)^d}\sigma^d} + \frac{d(c+r)^2}{2\sigma^2} =: c_2. \tag{12}$$

For any $(\mathbf{X}, Y)$ from the bGMM$(\boldsymbol{\mu}, \sigma)$ and any $\boldsymbol{\theta} \in \Theta_{\boldsymbol{\mu},c}$, the probability that $l(\boldsymbol{\theta}, (\mathbf{X}, Y))$ belongs to the interval $[c_1, c_2]$ ($c_1$, $c_2$ depend on $\delta_{r,d}$) can be lower bounded by

$$\Pr\left(l(\boldsymbol{\theta}, (\mathbf{X}, Y)) \in [c_1, c_2]\right) \geq 1 - \delta_{r,d}. \tag{13}$$

Thus, according to Hoeffding's lemma, with probability at least $1 - \delta_{r,d}$, $l(\boldsymbol{\theta}, (\mathbf{X}, Y)) \sim \mathsf{subG}((c_2 - c_1)/2)$ under $(\mathbf{X}, Y) \sim \mathcal{N}(Y\boldsymbol{\mu}, \sigma^2 \mathbf{I}_d) \otimes P_Y$ for all $\boldsymbol{\theta} \in \Theta_{\boldsymbol{\mu},c}$, i.e., for all $\lambda \in \mathbb{R}$,

$$\mathbb{E}_{\mathbf{X},Y}\left[\exp\left(\lambda\left(l(\boldsymbol{\theta}, (\mathbf{X}, Y)) - \mathbb{E}_{\mathbf{X},Y}[l(\boldsymbol{\theta}, (\mathbf{X}, Y))]\right)\right)\right] \leq \exp\left(\frac{\lambda^2 (c_2 - c_1)^2}{8}\right). \tag{14}$$

Under this setup, the iterative SSL procedure is shown in Figure 2, but the labelled dataset $S_l$ is only used to train in the initial round $t = 0$; we discuss the use of $S_l$ in all iterations in Corollary 3. The algorithm operates in the following steps.

- **Step 1: Initial round** $t = 0$ **with** $S_l$**:** By minimizing the empirical risk of labelled dataset $S_l$

$$L_{S_l}(\boldsymbol{\theta}) = \frac{1}{n} \sum_{i=1}^{n} l(\boldsymbol{\theta}, (\mathbf{X}_i, Y_i)) \overset{c}{=} \frac{1}{2\sigma^2 n} \sum_{i=1}^{n} (\mathbf{X}_i - Y_i \boldsymbol{\theta})^\top (\mathbf{X}_i - Y_i \boldsymbol{\theta}), \tag{15}$$

where $\overset{c}{=}$ means that both sides differ by a constant independent of $\boldsymbol{\theta}$, we obtain the minimizer

$$\boldsymbol{\theta}_0 = \arg\min_{\boldsymbol{\theta} \in \Theta} L_{S_l}(\boldsymbol{\theta}) = \frac{1}{n} \sum_{i=1}^{n} Y_i \mathbf{X}_i. \tag{16}$$

- **Step 2: Pseudo-label data in** $S_u$**:** At each iteration $t \in [1 : \tau]$, for any $i \in [(t-1)m + 1 : tm]$, we use $\boldsymbol{\theta}_{t-1}$ to assign a pseudo-label for $\mathbf{X}'_i$, that is, $\hat{Y}'_i = f_{\boldsymbol{\theta}_{t-1}}(\mathbf{X}'_i) = \mathrm{sgn}(\boldsymbol{\theta}_{t-1}^\top \mathbf{X}'_i)$.

- **Step 3: Refine the model:** We then use the pseudo-labelled dataset $\hat{S}_{u,t}$ to train the new model. By minimizing the empirical risk of $\hat{S}_{u,t}$

$$L_{\hat{S}_{u,t}}(\boldsymbol{\theta}) = \frac{1}{m} \sum_{i=(t-1)m+1}^{tm} l(\boldsymbol{\theta}, (\mathbf{X}'_i, \hat{Y}'_i)) \overset{c}{=} \frac{1}{2\sigma^2 m} \sum_{i=(t-1)m+1}^{tm} (\mathbf{X}'_i - \hat{Y}'_i \boldsymbol{\theta})^\top (\mathbf{X}'_i - \hat{Y}'_i \boldsymbol{\theta}), \tag{17}$$

we obtain the new model parameter

$$\boldsymbol{\theta}_t = \frac{1}{m} \sum_{i=(t-1)m+1}^{tm} \hat{Y}'_i \mathbf{X}'_i = \frac{1}{m} \sum_{i=(t-1)m+1}^{tm} \mathrm{sgn}(\boldsymbol{\theta}_{t-1}^\top \mathbf{X}'_i) \mathbf{X}'_i. \tag{18}$$

If $t < \tau$, go back to Step 2.

To state our result succinctly, we first define some non-standard notations and functions. From (16), we know that $\boldsymbol{\theta}_0 \sim \mathcal{N}(\boldsymbol{\mu}, \frac{\sigma^2}{n}\mathbf{I}_d)$ and inspired by Oymak & Gulcu (2021), we can decompose $\boldsymbol{\theta}_0$ as $\boldsymbol{\theta}_0 = (1 + \frac{\sigma}{\sqrt{n}}\xi_0)\boldsymbol{\mu} + \frac{\sigma}{\sqrt{n}}\boldsymbol{\mu}^\perp$, where $\xi_0 \sim \mathcal{N}(0, 1)$, $\boldsymbol{\mu}^\perp \sim \mathcal{N}(\mathbf{0}, \mathbf{I}_d - \boldsymbol{\mu}\boldsymbol{\mu}^\top)$, and $\boldsymbol{\mu}^\perp$ is perpendicular to $\boldsymbol{\mu}$ and independent of $\xi_0$ (the details of this decomposition are provided in Appendix C).

Given two vectors $(\mathbf{a}, \mathbf{b})$, define their correlation as $\rho(\mathbf{a}, \mathbf{b}) := \frac{\langle \mathbf{a}, \mathbf{b} \rangle}{\|\mathbf{a}\|_2 \|\mathbf{b}\|_2}$ in $[-1, 1]$. The correlation between the estimated parameter $\boldsymbol{\theta}_0$ and true parameter $\boldsymbol{\mu}$ is given by

$$\alpha(\xi_0, \boldsymbol{\mu}^\perp) := \rho(\boldsymbol{\theta}_0, \boldsymbol{\mu}) = \frac{1 + \frac{\sigma}{\sqrt{n}}\xi_0}{\sqrt{(1 + \frac{\sigma}{\sqrt{n}}\xi_0)^2 + \frac{\sigma^2}{n}\|\boldsymbol{\mu}^\perp\|_2^2}}. \tag{19}$$

Let $\beta(\xi_0, \boldsymbol{\mu}^\perp) = \sqrt{1 - \alpha(\xi_0, \boldsymbol{\mu}^\perp)^2}$. We abbreviate $\alpha(\xi_0, \boldsymbol{\mu}^\perp)$ and $\beta(\xi_0, \boldsymbol{\mu}^\perp)$ to $\alpha$ and $\beta$ respectively in the following. We can decompose the normalized vector $\boldsymbol{\theta}_0 / \|\boldsymbol{\theta}_0\|_2$ as follows

$$\bar{\boldsymbol{\theta}}_0 := \frac{\boldsymbol{\theta}_0}{\|\boldsymbol{\theta}_0\|_2} = \alpha\boldsymbol{\mu} + \beta\boldsymbol{v}, \tag{20}$$

where $\boldsymbol{v} = \boldsymbol{\mu}^{\perp}/\|\boldsymbol{\mu}^{\perp}\|_2$. Let $\bar{\boldsymbol{\theta}}_0^{\perp} := (2\beta^2\boldsymbol{\mu} - 2\alpha\beta\boldsymbol{v})/\sigma$, which is a vector perpendicular to $\bar{\boldsymbol{\theta}}_0$.

Define the *KL-divergence between the pseudo-labelled data distribution and the true data distribution after the first iteration* $G_\sigma : [-1,1] \times \mathbb{R} \times \mathbb{R}^d \to [0,\infty)$ as

$$G_\sigma(\alpha, \xi_0, \boldsymbol{\mu}^{\perp}) := D\left(\Phi\left(\frac{-\alpha}{\sigma}\right)p_{\tilde{g}+\frac{2\alpha}{\sigma}|\tilde{g}\leq\frac{-\alpha}{\sigma}} \otimes p_{\tilde{\mathbf{g}}^{\perp}+\bar{\boldsymbol{\theta}}_0^{\perp}} + \Phi\left(\frac{\alpha}{\sigma}\right)p_{\tilde{g}|\tilde{g}\leq\frac{\alpha}{\sigma}} \otimes p_{\tilde{\mathbf{g}}^{\perp}} \Big\| p_{\tilde{g}} \otimes p_{\tilde{\mathbf{g}}^{\perp}}\right), \quad (21)$$

where $\tilde{g} \sim \mathcal{N}(0,1)$, $\tilde{\mathbf{g}}^{\perp} \sim \mathcal{N}(0, \mathbf{I}_d - \bar{\boldsymbol{\theta}}_0\bar{\boldsymbol{\theta}}_0^{\top})$, $\tilde{\mathbf{g}}^{\perp}$ is independent of $\tilde{g}$ and perpendicular to $\bar{\boldsymbol{\theta}}_0$. Note that $p_{\tilde{g}+\frac{2\alpha}{\sigma}|\tilde{g}\leq\frac{-\alpha}{\sigma}}$ is the Gaussian probability density function with mean $\frac{2\alpha}{\sigma}$ and variance 1 *truncated* to the interval $(-\infty, -\frac{\alpha}{\sigma})$, and similarly for $p_{\tilde{g}|\tilde{g}\leq\frac{\alpha}{\sigma}}$. In general, when $G_\sigma(\alpha, \xi_0, \boldsymbol{\mu}^{\perp})$ is small, so is the generalization error.

Let $Q(\cdot) := 1 - \Phi(\cdot)$. Define the *correlation evolution function* $F_\sigma : [-1,1] \to [-1,1]$ that quantifies the increase to the correlation (between the current model parameter and the optimal one) and improvement to the generalization error as the iteration counter increases from $t$ to $t+1$:

$$F_\sigma(x) := \left(1 + \frac{\frac{2\sigma^2(1-x^2)}{\pi}\exp(-\frac{x^2}{\sigma^2})}{\left(1 - 2Q(\frac{x}{\sigma}) + \frac{2\sigma x}{\sqrt{2\pi}}\exp(-\frac{x^2}{2\sigma^2})\right)^2}\right)^{-\frac{1}{2}}. \quad (22)$$

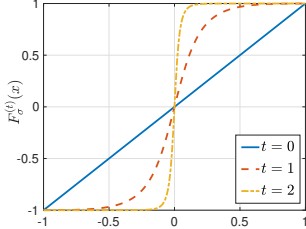

**Figure 3:** $F_\sigma^{(t)}(x)$ versus $x$ for different $t$ when $\sigma = 0.5$.

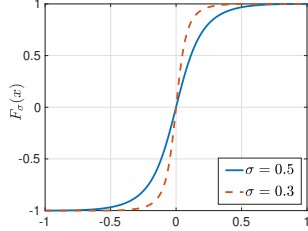

**Figure 4:** $F_\sigma(x)$ versus $x$ for $\sigma = 0.3$ and $0.5$.

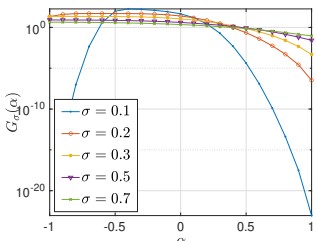

**Figure 5:** $G_\sigma(\alpha)$ versus $\alpha$ for different $\sigma$.

The $t^{\text{th}}$ iterate of the function $F_\sigma$ is defined as $F_\sigma^{(t)} := F_\sigma \circ F_\sigma^{(t-1)}$ with $F_\sigma^{(0)}(x) = x$. As shown in Figure 3, for any fixed $\sigma$, we can see that $F_\sigma^{(2)}(x) \geq F_\sigma(x) \geq x$ for $x \geq 0$ and $F_\sigma^{(2)}(x) < F_\sigma(x) < x$ for $x < 0$. It can also be easily deduced that for any $t \in [0:\tau]$, $F_\sigma^{(t+1)}(x) \geq F_\sigma^{(t)}(x)$ for any $x \geq 0$ and $F_\sigma^{(t+1)}(x) < F_\sigma^{(t)}(x)$ for any $x < 0$. This important observation implies that if the correlation $\alpha$, defined in (19), is positive, $F_\sigma^{(t)}(\alpha)$ increases with $t$; and vice versa. Moreover, as shown in Figure 4, by varying $\sigma$, we can see that smaller $\sigma$ results in a larger $|F_\sigma(x)|$.

By applying the result in Theorem 1, the following theorem provides an upper bound for the generalization error at each iteration $t$ for $m$ large enough.

**Theorem 2.** *Fix any* $\sigma \in \mathbb{R}_+$, $d \in \mathbb{N}$, $\epsilon \in \mathbb{R}_+$ *and* $\delta \in (0,1)$. *With probability at least* $1 - \delta$, *the absolute generalization error at* $t = 0$ *can be upper bounded as follows*

$$\left|\mathrm{gen}_0(P_{\mathbf{Z}}, P_{\mathbf{X}}, P_{\boldsymbol{\theta}_0|S_1,S_u})\right| \leq \sqrt{\frac{(c_2-c_1)^2 d}{4}\log\frac{n}{n-1}}. \quad (23)$$

*For each* $t \in [1:\tau]$, *for* $m$ *large enough, with probability at least* $1 - \delta$,

$$\left|\mathrm{gen}_t(P_{\mathbf{Z}}, P_{\mathbf{X}}, \{P_{\boldsymbol{\theta}_k|S_1,S_u}\}_{k=0}^t, \{f_{\boldsymbol{\theta}_k}\}_{k=0}^{t-1})\right|$$
$$\leq \sqrt{\frac{(c_2-c_1)^2}{2}} \mathbb{E}_{\xi_0,\boldsymbol{\mu}^{\perp}}\left[\sqrt{G_\sigma\left(F_\sigma^{(t-1)}(\alpha(\xi_0,\boldsymbol{\mu}^{\perp})),\xi_0,\boldsymbol{\mu}^{\perp}\right) + \epsilon}\right]. \quad (24)$$

The proof of Theorem 2 is provided in Appendix C. Several remarks are in order.

First, to gain more insight, we numerically plot $G_\sigma(\alpha, \xi_0, \boldsymbol{\mu}^{\perp})$ when $d = 2$ and $\boldsymbol{\mu} = (1,0)$ in Figure 5. Under these settings, $G_\sigma(\alpha, \xi_0, \boldsymbol{\mu}^{\perp})$ depends only on $\alpha$ and hence, we can rewrite it as $G_\sigma(\alpha)$. As shown in Figure 5, for all $\sigma_1 > \sigma_2$, there exists an $\alpha_0 \in [-1,1]$ such that for all

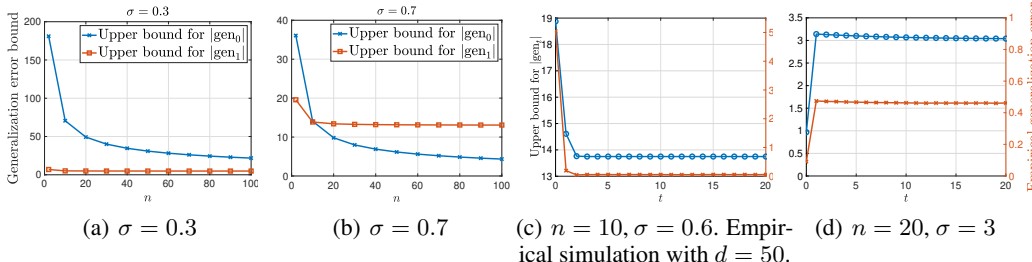

**Figure 6:** (a) and (b): Upper bounds for generalization error at $t = 0$ and $t = 1$ under different $\sigma$ when $d = 2$ and $\boldsymbol{\mu} = (1, 0)$. (c) and (d): The comparison between the upper bound for $|\text{gen}_t|$ and the empirical generalization error at each iteration $t$. The upper bounds are both for $d = 2$.

$\alpha \geq \alpha_0 = \alpha_0(\sigma_1, \sigma_2)$, $G_{\sigma_1}(\alpha) > G_{\sigma_2}(\alpha)$. From (19), we can see that $\alpha$ is close to 1 of high probability, which means that $\sigma \mapsto G_\sigma(\alpha)$ is monotonically increasing in $\sigma$ with high probability. As a result, $\mathbb{E}_\alpha[\sqrt{G_\sigma(\alpha)}]$ increases as $\sigma$ increases. This is consistent with the intuition that when the training data has larger variance, it is more difficult to generalize well. Moreover, for $\alpha > 0$, $G_\sigma(\alpha)$ decreases as $\alpha$ increases. Since $F_\sigma^{(t)}(\alpha)$ is increasing in $t$ for $\alpha > 0$, then $G_\sigma(F_\sigma^{(t)}(\alpha))$ is decreasing in $t$, which implies that the upper bound in (24) is also decreasing in $t$.

**Remark 1.** *As $n \to \infty$, $\boldsymbol{\theta}_0 \to \boldsymbol{\mu}$ and $\alpha = \rho(\boldsymbol{\theta}_0, \boldsymbol{\mu}) \to 1$ almost surely, which means that the estimator converges to the optimal classifier for this bGMM. However, since there is no margin between two groups of data samples, the error probability $\Pr(\hat{Y}'_j \neq Y'_j) \to Q(1/\sigma) > 0$ (which is the Bayes error rate) and the disintegrated KL-divergence $D_{\xi_0, \boldsymbol{\mu}^\perp}(P_{\mathbf{X}'_j, \hat{Y}'_j} \| P_{\mathbf{X}, Y})$ between the estimated and underlying distributions cannot converge to 0. We discuss the other extreme case in which $\alpha = -1$ in Remark 2 in Appendix C of the supplementary material.*

Second, by letting $\epsilon \to 0$, we compare the upper bounds for $|\text{gen}_0|$ and $|\text{gen}_1|$, as shown in Figures 6(a) and 6(b). For any fixed $\sigma$, when $n$ is sufficiently small, the upper bound for $|\text{gen}_0|$ is greater than that for $|\text{gen}_1|$. As $n$ increases, the upper bound for $|\text{gen}_1|$ surpasses that of $|\text{gen}_0|$, as shown in Figure 6(b). This is consistent with the intuition that when the labelled data is limited, using the unlabelled data can help improve the generalization performance. However, as the number of labelled data increases, using the unlabelled data may degrade the generalization performance, if the distributions corresponding to classes $+1$ and $-1$ have a large overlap. This is because the labelled data is already effective in learning the unknown parameter $\boldsymbol{\theta}_t$ well and additional pseudo-labelled data does not help to further boost the generalization performance. Furthermore, by comparing Figures 6(a) and 6(b), we can see that for smaller $\sigma$, the improvement from $|\text{gen}_0|$ to $|\text{gen}_1|$ is more pronounced. The intuition is that when $\sigma$ decreases, the data samples have smaller variance and thus the pseudo-labelling is more accurate. In this case, unlabelled data can improve the generalization performance. Let us examine the effect of $n$, the number of labelled training samples. By expanding $\alpha$, defined in (19), using a Taylor series, we have

$$\alpha = 1 - \frac{\sigma^2}{2n} \|\boldsymbol{\mu}^\perp\|_2^2 + o\left(\frac{1}{n}\right). \tag{25}$$

It can be seen that as $n$ increases, $\alpha$ converges to 1 in probability. Suppose the dimension $d = 2$ and $\boldsymbol{\mu} = (1, 0)$. Then $\boldsymbol{\mu}^\perp = [0, \mu_2^\perp]$ where $\mu_2^\perp \sim \mathcal{N}(0, 1)$. The upper bound for the absolute generalization error at $t = 1$ can be rewritten as

$$|\text{gen}_1| \lessapprox \sqrt{\frac{(c_2 - c_1)^2}{2}} \int_{-\sqrt{2}}^{\sqrt{2}} \frac{\sqrt{n}}{\sqrt{\pi}\sigma} e^{-\frac{ny^2}{\sigma^2}} \sqrt{G_\sigma(1 - y^2)} \, dy, \tag{26}$$

which is a decreasing function of $n$, as shown in Figures 6(a) and 6(b).

Third, given any pair of $(\xi_0, \boldsymbol{\mu}^\perp)$, if $\alpha(\xi_0, \boldsymbol{\mu}^\perp) > 0$, $F_\sigma^{(t)}(\alpha(\xi_0, \boldsymbol{\mu}^\perp)) > F_\sigma^{(t-1)}(\alpha(\xi_0, \boldsymbol{\mu}^\perp))$ for all $t \in [1 : \tau]$, as shown in Figure 3. This means that if the quality of the labelled data $S_1$ is reasonably good, by using $\boldsymbol{\theta}_0$ which is learned from $S_1$, the generated pseudo-labels for the unlabelled data are largely correct. Then the subsequent parameters $\boldsymbol{\theta}_t, t \geq 1$ learned from the large number of pseudo-labelled data examples can improve the generalization error. Therefore, the upper bound for $|\text{gen}_t|$ decreases as $t$ increases. In Figure 6(c), we plot the theoretical upper bound in (24) by

ignoring $\epsilon$. Unfortunately it is computationally difficult to numerically calculate the bound in (24) for high dimensions $d$ (due to the need for high-dimensional numerical integration), but we can still gain insight from the result for $d = 2$. It is shown that the upper bound for $|\mathrm{gen}_t|$ decreases as $t$ increases and finally converges to a non-zero constant. The gap between the upper bounds for $|\mathrm{gen}_t|$ and for $|\mathrm{gen}_{t+1}|$ decreases as $t$ increases and shrinks to almost 0 for $t \geq 2$. The intuition is that as $m \to \infty$, there are sufficient data at each iteration and the algorithm can converge at very early stage. In the empirical simulation, we let $d = 50$, $\boldsymbol{\mu} = (1, 0, \ldots, 0)$ and iteratively run the self-training procedure for 20 iterations and 2000 rounds. We find that the behaviour of the empirical generalization error (the red '-x' line) is similar to the theoretical upper bound (the blue '-o' line), which almost converges to its final value at $t = 2$. This result shows that the theoretical upper bound in (24) serves as a useful rule-of-thumb for how the generalization error changes over iterations. In Figure 6(d), we plot the theoretical bound and result from the empirical simulation based on the toy example for $d = 2$ but larger $n$ and $\sigma$. This figure shows that when we increase $n$ and $\sigma$, using unlabelled data may not be able to improve the generalization performance. The intuition is that for $n$ large enough, merely using the labelled data can yield sufficiently low generalization error and for subsequent iterations with the pseudo-labelled data, the reduction in the test loss is negligible but the training loss will decrease more significantly (thus causing the generalization error to increase). When $\sigma$ is larger, the data samples have larger variance and the classes have a larger overlap, and thus, the initial parameter $\boldsymbol{\theta}_0$ learned by the labelled data cannot produce pseudo-labels with sufficiently high accuracy. Thus, the pseudo-labelled data cannot help to improve the generalization error significantly.

Fourth, we consider an "enhanced" scenario in which the labelled data in $S_1$ are reused in each iteration. Set $w = \frac{n}{n+m}$ in (5). We can extend Theorem 2 to Corollary 3 as follows. Similarly to $F_\sigma$, let us define the *enhanced correlation evolution function* $\tilde{F}_{\sigma, \xi_0, \boldsymbol{\mu}^\perp} : [-1, 1] \to [-1, 1]$ as follows:

$$\tilde{F}_{\sigma, \xi_0, \boldsymbol{\mu}^\perp}(x) = \left(1 + \frac{\left(w\frac{\sigma\|\boldsymbol{\mu}^\perp\|_2}{n} + (1-w)(\frac{2\sigma\sqrt{1-x^2}}{\sqrt{2\pi}}\exp(-\frac{x^2}{2\sigma^2}))\right)^2}{\left(w(1+\frac{\sigma}{\sqrt{n}}\xi_0) + (1-w)(1 - 2Q(\frac{x}{\sigma}) + \frac{2\sigma x}{\sqrt{2\pi}}\exp(-\frac{x^2}{2\sigma^2}))\right)^2}\right)^{-\frac{1}{2}}. \quad (27)$$

**Corollary 3.** *Fix any $\sigma \in \mathbb{R}_+$, $d \in \mathbb{N}$, $\epsilon \in \mathbb{R}_+$ and $\delta \in (0, 1)$. For $m$ large enough, with probability at least $1 - \delta$, the absolute generalization error at any $t \in [1 : \tau]$ can be upper bounded as follows*

$$\left|\mathrm{gen}_t(P_{\mathbf{Z}}, P_{\mathbf{X}}, \{P_{\boldsymbol{\theta}_k|S_1, S_\mathrm{u}}\}_{k=0}^t, \{f_{\boldsymbol{\theta}_k}\}_{k=0}^{t-1})\right| \leq w\sqrt{\frac{(c_2-c_1)^2 d}{4}\log\frac{n}{n-1}}$$

$$+ (1-w)\sqrt{\frac{(c_2-c_1)^2}{2}}\,\mathbb{E}_{\xi_0, \boldsymbol{\mu}^\perp}\left[\sqrt{G_\sigma\big(\tilde{F}_{\sigma, \xi_0, \boldsymbol{\mu}^\perp}^{(t-1)}(\alpha(\xi_0, \boldsymbol{\mu}^\perp)), \xi_0, \boldsymbol{\mu}^\perp\big) + \epsilon}\right]. \quad (28)$$

The details are provided in Appendix D and the proof of Corollary 3 is provided in Appendix E. It can be seen from Figure 11 that the new upper bound for $|\mathrm{gen}_t|$ remains as a decreasing function of $t$. We find that when $n = 10$, $m = 1000$, the upper bound is almost the same as that one in Figure 6(c), which means that for large enough $\frac{m}{n}$, reusing the labelled data does not necessarily help to improve the generalization performance. Moreover, when $m = 100$, the upper bound is higher than that for $m = 1000$, which coincides with the intuition that increasing the number of unlabelled data helps to reduce the generalization error.

## 5 EXPERIMENTAL RESULTS

In Sections 3 and 4, we theoretically analyse the upper bound of generalization error across the iterations for iterative self-training and especially for the case of bGMM classification. In this section, we conduct experiments on real datasets to demonstrate that our theoretical results on the bGMM example can also reflect the training dynamics on complicated tasks.

We train deep neural networks via a iterative self-learning strategy (under the same setting as that for Corollary 3) to perform binary and multi-class classification tasks. In the first iteration, we only use the labelled data to optimize the deep neural network (DNN) and train the model for a relatively large number of epochs so that the training loss will converge to a small value and the model is initialized well. In the following iterations, we first sample a subset of unlabelled data from the whole set and generate pseudo-labels for them via the model trained in the previous iteration. Then, we update the model for a small number of epochs with both the labelled and pseudo-labelled data.

**Experimental settings:** For binary classification, we collect pairs of classes of images, i.e., "automobile" and "truck", "horse" and "ship", from the CIFAR10 (Krizhevsky, 2009) dataset. In this dataset, each class has 5000 images for training and 1000 images for testing. We use the whole set of images in the selected pair of categories and divide them into two sets, i.e., the labelled training set with 500 images and the unlabelled training set with 9500 images. We train a convolutional neural network, ResNet-10 (He et al., 2016), to minimize the cross-entropy loss via the self-learning strategy to perform the binary classification. The model is trained for 100 epochs in the first iteration and 20 epochs in the following iterations; we use the Adam (Kingma & Ba, 2015) optimizer with a learning rate of 0.001. In each iteration after the initial one, we sample 2500 unlabelled images assign them pseudo-labels. The complete training procedure lasts for 100 self-training iterations.

We further validate our theoretical contributions on a multi-class classification problem in which we train a ResNet-6 model with the cross-entropy loss to perform 10-class handwritten digits classification on the MNIST (LeCun et al., 1998) dataset. We sample 51000 images from the training set, which contains 6000 images for each of the ten classes. We divide them into two sets, i.e., a labelled training set with 1000 images and an unlabelled set with 50000 images. The optimizer and training iterations follow those in the aforementioned binary classification tasks.

**Experimental observations:** We perform each experiment 3 times and report the average test and training (cross entropy) losses, the generalization error, and test and training accuracies in Figures 7–9. The generalization error appears to have relatively large reduction in the early training iterations and then fluctuates around a constant value afterwards. For example, in Figure 7, the generalization error converges to around 0.25 after 30 iterations; in Figure 8, it converges to around 0.4 after 10 iterations; in Figure 9, it converges to around 0.1 after 12 iterations. These results corroborate the theoretical and empirical analyses in the bGMM case, which again verifies the validity of the proposed generalization error bound in Theorem 2 and Corollary 3 on benchmark datasets. It also reveals that the generalization performance of iterative self-training on real datasets from relatively distinguishable classes can be quickly improved with the help of unlabelled data. We also show that the test accuracy increases with the iterations and has significant improvement compared to the initial iteration when only labelled data are used. In Figure 7, the highest accuracy has about a 4% increase from the initial point; in Figure 8, there is about a 10% increase; and in Figure 9, there is about a 3% increase. Thus, these numerical results suggest that via iterative self-training with pseudo-labelling, not only can we improve the generalization error as the iteration count increases, but we can also enhance the test accuracy. In addition, apart from the "horse-ship" and "automobile-truck" pairs (that are relatively easy to distinguish based on the high classification accuracy and low loss as shown in Figures 7 and 8), we also perform another experiment (detailed in Appendix F) on a harder-to-distinguish pair, "cat" and "dog" (see Table 1), whose results show that the generalization error does not decrease with the iterations even though the classification accuracy increases. This again corroborates the results in Figure 6(d) for the bGMM with large variance.

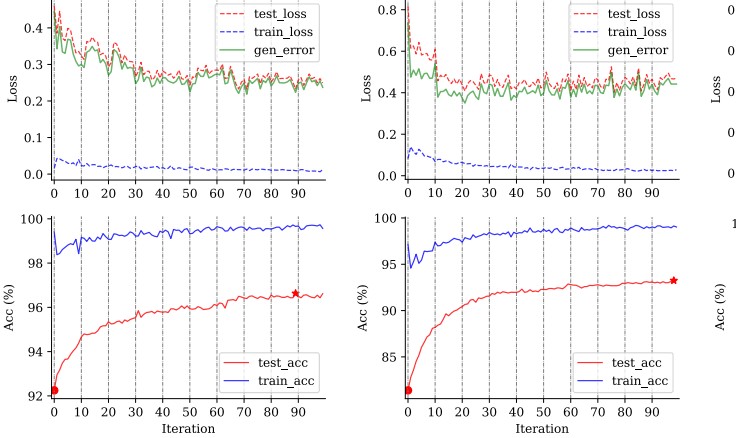
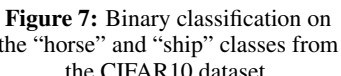
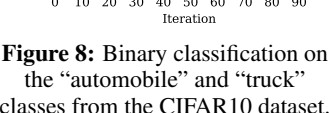
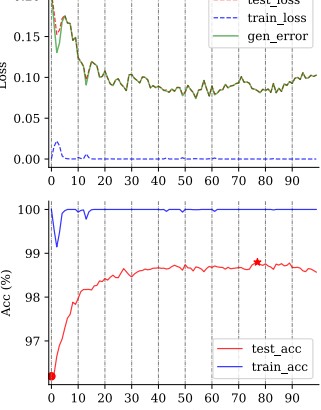

**Figure 7:** Binary classification on the "horse" and "ship" classes from the CIFAR10 dataset.

**Figure 8:** Binary classification on the "automobile" and "truck" classes from the CIFAR10 dataset.

**Figure 9:** 10-class classification on the MNIST handwritten digits dataset.

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
