# OpenReview forum: "Information-Theoretic Generalization Bounds for Iterative Semi-Supervised Learning"
_ICLR.cc/2022/Conference — ICLR 2022 Submitted_

### Official Review · Reviewer_hqP3 · 2021-10-31

**Correctness:** 3
**Technical Novelty And Significance:** 3
**Empirical Novelty And Significance:** 2
**Recommendation:** 6
**Confidence:** 4

**Main Review:**

I have a question regarding to the SSL algorithm considered in the paper, splitting of the unlabeled dataset into \tau disjoint part is something unnatural to me. I understand that making \hat{S}_{u,k}  independent would simplify the analysis, but can you give more discussion on such a formulation compared to using the entire unlabeled dataset and update the pseudo-label in each iteration? Can we interpret it as a bias-variance trade-off, as more samples would reduce variance, but the low quality of the pseudo-label would increase the bias?

The organization of section 4 can be improved by using subsections and subtitles, which emphasize the main take-away message of each discussion and remark in words. There are so many new notations introduced in this section, and they are not well-explained, especially the G_\sigma and F_\sigma function. If we only care about the monotonicity of these functions, then a lot of technical details can be moved to the appendix, so that the discussion of related works can be incorporated in the main body of the paper.

I feel like the experimental result is not very informative, as it only shows that similar generalization behavior (decrease and then saturate) can be observed in practice. However, such a phenomenon is expected, and something more useful would be how to use such bounds to predict the performance of SSL algorithm and further improve it. Maybe the authors could provide more discussion on the “cat” and “dog” result and talk about the influence of the label noise in real dataset.


**Summary Of The Paper:**

This paper provides a generalization error bound for iterative semi-supervised learning (SSL) algorithms using information-theoretic principles (see Theorem 1). To provide more intuitions, the authors first work with a simple model, i.e., the binary Gaussian mixture model (bGMM). It is shown in bGMM that when the class conditional variances are not too large, the upper bound on the generalization error decreases monotonically with the number of iterations, but quickly saturates. The theoretical results on the simple model are corroborated by experiments on MNIST and CIFAR datasets, where similar phenomena are observed, i.e., the generalization error improves after several pseudo-labelling iterations, but saturates afterwards.

**Summary Of The Review:**

This paper provides a novel information-theoretic generalization bound for semi-supervised learning, which involves both disintegrated mutual information (characterizing the input output sensitivity) and the disintegrated KL-divergence (distribution mismatch for pseudo-label) conditioned on the previous outputs. To my knowledge, the bGMM results are quite novel and interesting.

---

> ### Author Response · Authors · 2021-11-14
> **Response to Reviewer hqP3 [Part I]**
>
> We thank the reviewer for your meticulous reading and detailed comments. We have carefully addressed your concerns in the following.
>
> **Q1**:
> + Iterative pseudo-labeling is a widely used practical algorithm in SSL (Zhu \& Goldberg, 2009, Triguero et al., 2015), where one can either reuse or refresh the unlabeled datasets. However, as discussed in works by Arazo et al. (2020) and Oymak \& Gulcu (2020), reusing the unlabeled datasets may degrade the performance of the algorithms. Thus, in this paper we consider the setup of splitting the unlabeled datasets into disjoint parts.
>
> If we consider using all $\tau m$ unlabeled data in each iteration and iteratively update the pseudo-labels, the formulations will be changed to
> $$\text{The generalization error: } $$
> \begin{align}
> 	&\mathrm{gen}\_t =w\bigg(\mathbb{E}\_{\theta_t}[\mathbb{E}\_{Z}[l(\theta_t,Z)\mid \theta_t]]-\frac{1}{n}\sum_{i=1}^n \mathbb{E}\_{\theta_t,Z_i}[l(\theta_t,Z_i)] \bigg)
> 	 +(1-w)\bigg(\mathbb{E}\_{\theta_t}[\mathbb{E}\_{Z}[l(\theta_t,Z)\mid \theta_t]]-\textcolor{blue}{\frac{1}{\tau m}\sum_{i=1}^{\tau m}\mathbb{E}\_{\theta_t,X'_i,\hat{Y}'_i}[l(\theta_t,(X'_i,\hat{Y}'_i))]} \bigg).
> \end{align}
> In Eq. (8) of Theorem 1, the empirical average taken for the losses in $[(t-1) m+1:tm]$ will be changed to $[1:\tau m]$. Up to this point, there is no essential change to the formulations but we need to notice that each $\mathbf{\theta}\_t$ for $t\geq 1$ will be dependent on all the pseudo-labeled data.
>
> Next, when we consider the specific example of the bGMM classification problem, in Step 3 of the algorithm (Eqs. (17) and (18)), we will need to include all the unlabeled data samples and the updated parameter $\mathbf{\theta}\_t$ will be dependent on all $X_i'$ in $S_u$, which will induce greater complexity in calculating the mutual information between $\mathbf{\theta}\_t$ and $(\mathbf{X}\_i',\hat{Y}\_i')$ as well as the pseudo-labeling distribution $P_{\mathbf{X}\_i',\hat{Y}\_i'}$. To make it more computable and gain clearer insight, we instead assume that at each iteration, we use a fresh subset of the unlabeled data for pseudo-labeling to update $\mathbf{\theta}\_t$. Intuitively, reusing the whole unlabeled dataset $S_u$ may not help to decrease the generalization error as much as using a new subset $S_{u,t}$ at each $t$, since retraining tends to lead to overfitting. It would be an interesting direction in the future to further investigate.
>
> +  Actually the variance defined in our paper is a parameter in the data distributions and not affected by the number of samples, different from the variance in the traditional bias-variance tradeoff. In bGMM, larger variance means larger overlapping between two classes of data samples. Thus, when the variance is too large, the initialization based on the limited number of labeled samples will be quite bad and the quality of pseudo-labels will also be bad such that the generalization error even increases. Thus, we believe this is not due to the traditional bias-variance tradeoff.
>
> **Q2**: We thank the reviewer for the suggestion. We will reorganize Section 4 into several subsections for clarity if the paper is accepted.
>
> **Q3**: We respectfully disagree.
> 1. Although many experimental papers have empirically shown that self-training with pseudo-labelling works, there is no theoretical guarantee. In our paper, we managed to provide an theoretical bound from an information-theoretic viewpoint.
>
> 2. What our theoretical result informs us is that the variance of the classes $\sigma^2$ or their overlap is critical in the behavior of the evolution of the generalization error in iterative SSL algorithms. If $\sigma^2$ is small (resp.\ large), the generalization error improves (resp.\ worsens) as $t$ grows. This is also validated in our experiment on "cat-dog" datasets, which are analogous to bGMM with large $\sigma^2$. Based on our upper bound,
> there is a "sweet spot" for $\sigma^2$ as we show through an example of bGMM in the following table. We can numerically find the "sweet spot" for $\sigma^2$ (in-between $0.7^2$ and $0.8^2$) such that for any $\sigma^2$ smaller than this point,  pseudo-labeling can help to improve the generalization error; see the table below.
>
> **Table**: The gap between the upper bound for $|\mathrm{gen}_0|$ and that for $|\mathrm{gen}_1|$  under different $\sigma$ when $d=2$ and $n=10$. Positive values mean the generalization error bound is improved.
>
> | $\sigma$  | 0.4 | 0.5 | 0.6 | 0.7 | 0.8 | 0.9 | 1 | 1.1 | 1.2  |
> |:----:|:----:|:----:|:----:|:----:|:----:|:----:|:----:|:----:|:----:|
> | $n=10:$  | $29.6048$ | $12.5420$ | $4.2634$ | $\textcolor{red}{0.1992}$ | $\textcolor{red}{-1.8089}$ | $-2.7896$ | $-3.2439$ | $-3.4220$ | $-3.4530$ |
>
> **[Part II continued in the following response]**

---

> > ### Author Response · Authors · 2021-11-14
> > **Response to Reviewer hqP3 [Part II]**
> >
> > **Q3**:
> >
> > 3. We do not think there is direct connection between our results and the label noise in real datasets. The given labeled data from CIFAR10 are clean without noise in our setting. Although the pseudo-labeling may generate incorrect labels for a small subset of unlabeled data, the resultant label noise depends on the model learned from the previous iteration and we can derive the generalization error bound in a recursive way. However, the label noise in real dataset is completely random. We cannot assess the influence of the label noise unless certain assumptions are made, e.g., labels are corrupted by a fixed transition matrix. However, this is clearly out of scope of the analyses in our paper.
> >
> > 4. In our extended work, we find that our theoretical framework can also be used to explain the effect of threshold in pseudo-labeling. When the threshold is large, the benefit of pseudo-labeling will be degraded. This part of the work will be added to the main paper in the future.
> >
> > ----
> > **References**
> >
> > Eric Arazo, Diego Ortego, Paul Albert, Noel E O’Connor, and Kevin McGuinness. Pseudo-labeling
> > and confirmation bias in deep semi-supervised learning. In 2020 International Joint Conference on Neural Networks (IJCNN), 2020.

---

### Official Review · Reviewer_7Xsu · 2021-11-03

**Correctness:** 3
**Technical Novelty And Significance:** 2
**Empirical Novelty And Significance:** 1
**Recommendation:** 5
**Confidence:** 5

**Main Review:**

**Strength**

The paper presents a new general algorithm-dependent generalization bound for semi-supervised learning with iterative pseudo-labelling (Theorem 1) and an application of the bound to binary classification with Gaussian class conditionals (Theorem 2).

**Weakness**

Although the work is interesting and has some merit, to this reviewer, the theoretical results (Theorems 1 and 2) are quite far from the setting in the real world in which deep neural nets are used as the classifier and the training is done via SGD. Specifically

* Theorem 2 considers only a toy problem setting. In fact this setting is very well studied in Castelli and Cover (1996) ("The Relative Value of Labeled and Unlabeled Samples in Pattern Recognition with an Unknown Mixing Parameter") albeit not considering an iterative pseudo-labeling approach. Compared with Castelli and Cover (1996), the current paper does not seem to offer deeper insight or stronger results. -- Agreeably it may not be fair to compare the two papers, since the considered learning algorithms are different. But I invite the authors to study this classical paper on semi-supervised learning, which is missed in the reference list of this paper.

* Theorem 1 does have the potential for analyzing SGD-alike learning algorithms on deep neural nets. But the authors did not pursue further along this direction, while only studying such more practical settings via experiments.

The experimental results related to deep neural networks appear distant from the theoretical development. To this reviewer, they are inadequate to validate or support the theoretical development.


**Summary Of The Paper:**

The paper considers the problem of semi-supervised learning where pseudo-labeling is used to iteratively assign labels for unlabelled data batches to enlarge the labelled dataset for subsequent re-training of the classification model. The paper first adapts the recent results of Bu et al (2020) and Wu et al (2020) to this set-up and provides a general information-theoretic upper bound for the generalization error of such a learning algorithm. The paper then specializes its set-up to a binary classification problem with Gaussian class conditionals and presents its corresponding generalization bound. Additional experiments are performed on the more practical datasets with deep neural network classifiers.

**Summary Of The Review:**

The paper presents new generalization bounds for semi-supervised learning under iterative pseudo-labeling. But the results are distant from settings of practical interest.

---

> ### Author Response · Authors · 2021-11-14
> **Response to Reviewer 7Xsu**
>
> We thank the reviewer for your meticulous reading and useful comments. We have carefully addressed your concerns in the following.
>
> **Q1**: Thanks for pointing us to the classical paper by Castelli and Cover (1996) ("The Relative Value of Labeled and Unlabeled Samples in Pattern Recognition with an Unknown Mixing Parameter"). The authors of this paper studied the classification problem in a binary mixture model with known (but anonymous in the second setting) conditional distributions and unknown mixing parameter, where the optimal classifier is known as the Bayes classifier.
> They characterized the relative value of labeled and unlabeled data in improving the convergence rate of classification error probability. This paper is indeed an interesting work, which can give us insight about combining the labeled and unlabeled data during learning. We will incorporate the paper into our references if the paper is accepted.
>
> However, their problem setup assumes *known* distributions and is based on the likelihood ratio test, while in the real world we only have the training samples and typically, empirical risk minimization (ERM) is performed. In our paper, although we considered the simple bGMM classification problem, we actually considered  ERM and analysed the expected generalization error. Thus, our work  mimics reality in which the underlying distributions are unknown.
>
> **Q2**: We thank the reviewer for making this valid point. Indeed, Theorem 1 can be applied to analyzing algorithms like SGD, SGLD, noisy momentum and so on (Pensia et al. 2018, Negrea et al. 2019, Bu et al. 2020, Wu et al. 2020). This could be an interesting extension in the future work. However, our focus in this paper is not on analyzing the generalization error bound for different stochastic gradient algorithms but on how the generalization error bound evolves during the semi-supervised learning (SSL) with iterative pseudo-labelling method. Thus, we take a different direction after the Theorem 1. Other works particularize it to stochastic optimization algorithms. On the other hand, we particularize it to the iterative SSL framework, an orthogonal direction.
>
> **Q3**: As shown in Figures 6(c) and 6(d), we did empirical experiments on data from bGMM and compared the empirical generalization error with our derived theoretical upper bounds. It shows that the upper bound has the same behaviour with the empirical result as the iteration increases. However, we want to further validate that in real datasets, there will be similar phenomena as those in bGMM. As shown in Figures 7, 8, 9, and Table 1 and Figure 12 in Appendix F, for classes that are easier to classify (analogous to bGMM with small variance), the generalization error decreases and then saturates quickly. For classes that are difficult to distinguish (e.g., "cat-dog", analogous to bGMM with large variance), the generalization error does not decrease via iterative pseudo-labelling the unlabeled data. These experimental results actually correspond to our results on bGMM, which implies our upper bounds in Theorems 1 and 2, for the bGMM toy example, do have practical value in semi-supervised learning.
>
> We hope that the clarifications above are satisfactory and sincerely hope that the esteemed reviewer is open to upping the score.

---

> > ### Comment · Reviewer_7Xsu · 2021-11-15
> > **Follow-up**
> >
> > Thanks for the response.
> >
> > Regarding directions following Theorem 1, perhaps I did not make myself clear enough. What I had in mind is extending/adapting the result to analyzing semi-supervised classification problems for more practical settings (such as for image classification datasets) in which a neural network trained and retrained with SGD is used as the classifier. That would be a more interesting and practically relevant direction.
> >
> > My main concern is that the analysis of the paper focuses on a specific toy task with a specific training/retraining algorithm. It is difficult to justify the main result as analyzing the iterative SSL **framework**.
> >
> > Regarding your experimental validation on MNIST and CIFAR, if the main observation here is that the generalization error first improves with the number of unlabelled examples and then saturates, this result seems quite obvious: without including more labelled data, the role that unlabelled data may play is limited. Even in Castelli and Cover (1996), such a behaviour can be deduced from their Theorem 1 and Theorem 2. -- It is rightful to argue that Castelli and Cover (1996) has a different setting from the setup of your paper, but on the other hand, it is also rightful to argue that your experiments on MNIST and CIFAR also have a different set up from your theoretical analysis (your Theorem 2). That is, to what extent these experiments suggest a generalization of your Theorem 2 to a more practical setting is questionable -- why do they not suggest a generalization of the known results of Castelli and Cover (1996)? In any case, I feel the value of these experiments is limited.
> >
> > Nonetheless, I do see some merit in this work. But as explained above,  there seems no strong basis for me to raise my evaluation score.

---

> > > ### Author Response · Authors · 2021-11-16
> > > **Response to the follow-up**
> > >
> > > We thank the reviewer for the prompt response to ours.
> > >
> > > We submit that we *analyze* a toy example of a binary GMM. The reason for this is that any other classification model would result in analysis that, with high likelihood, yields a generalization error expression that is not amenable to interpretation and computation (hence of limited utility). The simple bGMM model informs us about the behavior of the generalization error for more realistic datasets, though we submit that this is perhaps a generous extrapolation.
> > >
> > > (*) However, we want to clarify that in the paper by Castelli and Cover (1996), they studied the probability of classification error, which is fundamentally different from the generalization error (i.e., the gap between the training and test losses). Thus, in our opinion, we respectfully disagree that the our experimental results on generalization error can be deduced from their paper.
> > > With regard to the reviewer's comment that it is obvious that "the generalization error first improves with the number of unlabelled examples and then saturates, this result seems quite obvious", we respectfully beg to differ. Note that this phenomenon occurs in the high signal-to-noise-ratio (SNR) regime. In the low SNR regime, this phenomenon no longer occurs, both theoretically and empirically. Rather the generalization error directly shoots up with iteration count as predicted theoretically and on the difficult cat-vs-dog classification example.
> > >
> > > We respect the reviewer's opinion, thank the reviewer for leaning towards an acceptance decision, but would just like to clarify the point made in paragraph (*).
> > >
> > > Regards,
> > >
> > > The authors

---

### Official Review · Reviewer_8ena · 2021-11-08

**Correctness:** 4
**Technical Novelty And Significance:** 2
**Empirical Novelty And Significance:** 2
**Recommendation:** 5
**Confidence:** 3

**Main Review:**

Strength: The conclusions and observations about generalization error is made in the regime where n -> infinity and m->infinity and m >>n. The analysis seems sound and credible. I did not check the proof in the appendix, but the results and overall proof sketch in the main paper seems reasonable.
Weakness: While the result is sound and theoretically valuable, it is not surprising at all in the regime studied (m and n large). The paper starts with the motivation that labeled data is limited, and pseudo labeling is shown to be helpful, but studying the regime where we have growing number of labeled samples does not completely intersect with that motivation. Also, the authors mention that after a few iterations, the accuracy might improve, but the generalization error does not, and I think it is worth elaborating on this point to see where this improvement is coming from.

Weakness: In the last section, the authors show with experiments on MNST and CIFAR datasets that the generalization error only decreases for a few iterations of pseudo labeling and then saturates. However, this observation is not very obvious to me by just looking at figures. Based on Figures 7, 8, and 9, there is a sharp reduction in generalization error after only one iteration, but after that, it is not necessarily decreasing. The overall trend of vanishing generalization error until a saturation limit is acceptable based on the plots, but drawing conclusions that for the first iterations we have significant reduction and then the reduction vanishes is not quite clear to me. For example, in figure 9, we have reduction in generalization error for the first iteration, then we have a jump in the generalization error followed by another sharp decline. It would be helpful if authors could explain these ups and downs in the plots.

Weakness: Also, the first iteration used a neural network that is well trained and converged to a good solution, then we use this well-trained network to predict labels for a subset of data, and fine-tune the network using the newly-labeled data. This is no surprise to me that more than a few rounds of fine-tuning will not improve the generalization error.

Strength: The paper is overall very well written and well-organized with the exception of section 4 where the authors could have done a better job of further dividing the section into multiple subsections.

Minor comments:

- In the informal statement of Theorem 1, it is better to mention that this bound is for pseudo labeling algorithm and not any SSL algorithm.
- In Equation 6, I could not find what f_{\theta_{k}} is.
- For section 4, it would be better to subdivide it into a few subsections. It was not smooth to follow the whole section.

**Summary Of The Paper:**

This paper considers one common semi-supervised learning algorithm, pseudo labeling, and studies this problem from theoretical point of view. Specifically, it derives an information theoretic upper bound on generalization error in each iterative update of pseudo labeling. They separate the bound into two main parts: one depends on the mutual information between the data samples and model parameters, and the other depends on the KL distance between the underlying data distribution and pseudo labeled samples from previous iteration. Their main conclusion is that as the number of labeled and unlabeled samples grows, the first term vanishes, but the second term does not necessarily vanish.

In the rest of the paper, the authors rely on the simple example of binary Gaussian Mixture Model to give a more understandable and sensible calculation of the their upper bound. Namely, they calculate the KL distance and mutual information terms in the main theorem and study the behavior of generalization error for this model. The conclusion they made is that the iterative pseudo labeling can decrease the generalization error for only the first few iterations and after than has no effect on reducing the generalization error.

**Summary Of The Review:**

The analysis of the paper seems interesting and correct. It is novel to look at the generalization error at each iterative update. However, I do have concerns about the significance of the work. It is not clear to me and it was not made clear by the authors how we can make use of this analysis and conclusions to train better models for example.

---

> ### Author Response · Authors · 2021-11-14
> **Response to Reviewer 8ena**
>
> We thank the reviewer for your careful reading and detailed comments.
>
> **Q1**:
> + We respectfully disagree with the reviewer. First, our result is only based on large $m$ but *finite* $n$; this is not unreasonable in practice because the number of labelled data $n$ is small but unlabelled data is aplenty. Our result is more than showing that pseudo-labeling improves the generalization error. In fact, it is highly sensitive to the noise variance (or overlap) $\sigma^2$ of the classes. As shown in Figures 6(c) and 6(d), if $\sigma^2$ is small, the generalization error behaves in the expected manner; it decreases with $t$. If $\sigma^2$ is larger, it in fact increases with $t$.
>
> + For the large-scale experiments, in   Table 1 in Appendix F, we quantify the difficulty of classifying different pairs of classes from  the CIFAR10 dataset and show that "horse-ship" and "automobile-truck" are relatively easy-to-distinguish classes (analogous to bGMM with small $\sigma^2$). That's why the results in Figures 7 and 8 are not surprising. However,  for "cat" and "dog" classes, they are much harder to distinguish (analogous to bGMM with large $\sigma^2$). As shown in Figure 12, pseudo-labeling the unlabeled data does not help to decrease the generalization error, even though the test accuracy increases. This result corroborates that for the bGMM. The fact that both the  generalization error and test accuracy appear to increase with $t$ is, in fact, not contradictory.
> We have provided a detailed  explanation in the third paragraph of Appendix F.
>
> **Q2**: In the large-scale experiments on real datasets, there is much randomness in the data so the samples do not necessarily conform to "nice" distributions. As such, the empirical generalization error does not monotonically decrease in each iteration. However, it can be seen in Figures 7, 8 and 9 that the *trend* of the generalization error is decreasing in the early stages and then plateaus.
>
> **Q3**: The reviewer's intuition is correct. However, our work formalizes this intuition from the perspective of information-theoretic generalization bounds. To the best of our knowledge, this has not been shown formally. Further, in Figures 6(d) and 12, we show that the improvement in the generalization error doesn't always occur, e.g., when the classification task is difficult.
>
> We hope that the clarifications above are satisfactory and sincerely hope that the esteemed reviewer is open to upping the score.

---

### Official Review · Reviewer_5JBw · 2021-11-08

**Correctness:** 3
**Technical Novelty And Significance:** 2
**Empirical Novelty And Significance:** 2
**Recommendation:** 5
**Confidence:** 3

**Main Review:**

Dear Authors,

Please find my review below. Sorry for submitting it late.

The reviewer likes the approach proposed in this paper; however, I would like to express several concerns regarding the theoretical power and limitations of the result. My main concern is about the results' impact and there how do they advance the state-of-the-art.  Please, find below some of the most crucial ones:
1. Theoretical bounds are based on the KL divergence between the pseudo-labeled and true distributions, which is hard to measure/estimate unless the additional information is specified. Any computable upper bound?
2. The loss function used for the generalization error computation is designed in such a way that even for a high misclassification rate on later stages the loss function would converge. (See Eq.~ 5 for the details)
3. No lower bound is provided. Any bound known from the literature?
4. Figure 1: the upper bound on the generalization error, which should be in between 0 and 1, is about 14 at convergence. Why is it so? A small number of objects? A similar concern regarding Figure 6.
5. When one is talking about the generalization error, one derives the bound for a class of functions rather than for a single function. However, in the paper

Overall, I believe the paper is interesting, but it can not be accepted as is unless the authors' resolve these comments.

**Summary Of The Paper:**

The paper considers a popular approach to semi-supervised learning based on iterative pseudo-labeling the unlabelled data and refining the model parameters thereafter. The paper is supported theoretically for the case of the Gaussian mixture model establishing a generalization error bound based on the KL divergence between the pseudo-labeled and true data distributions. The paper is also supported empirically for binary classification examples coming from CIFAR10 and MNIST datasets


**Summary Of The Review:**

The results are interesting; however, it is not clear to me how one can extend the results beyond the Gaussian mixture model considered in the paper. Furthermore, it is not clear how to get efficiently computable bounds from the statements of Theorems 1-3. The bounds are needed for model complexity penalization. Also, the following questions are needed to be addressed:

- Theoretical bounds are based on the KL divergence between the pseudo-labeled and true distributions, which is hard to measure/estimate unless the additional information is specified. Any computable upper bound?
- The loss function used for the generalization error computation is designed in such a way that even for a high misclassification rate on later stages the loss function would converge. (See Eq.~ 5 for the details). Could you advocate using this bound?
- No lower bound is provided. Any bound known from the literature?
- Figure 1: the upper bound on the generalization error, which should be in between 0 and 1, is about 14 at convergence. Why is it so? A small number of objects? A similar concern regarding Figure 6.
- When one is talking about the generalization error, one derives the bound for a class of functions rather than for a single function. However, in the paper

---

> ### Author Response · Authors · 2021-11-14
> **Response to Reviewer 5JBw [Part I]**
>
> We thank the reviewer for your meticulous reading and useful comments to improve our paper.
>
> **Q1**: Admittedly, the KL-divergence is hard to compute for general distributions. Hence, as to gain more insight, we adopt the simple but non-trivial bGMM as an example, under which the KL-divergence can be computed, as shown in Figures 6(c) and 6(c). We compare the theoretical upper bounds to the empirically obtained generalization error from synthetic experiments under different variances of the data distributions. The results show that the theoretical bound matches the behaviour of the real empirical generalization error as the iteration increases. Moreover, for small variance (easy classification problems), the generalization error quickly decreases and then saturates; for large variance (difficult classification problems), pseudo-labelling the unlabeled data does not help to decrease the generalization error.
>
> **Q2**: We apologize for not fully comprehending this comment so the following reply is based on our best understanding. The definition of the empirical risk in Eq.~(5) is widely used, following from the standard definitions in works like Shalev-Shwartz \& Ben-David (2014) and Xu \& Raginsky (2017).
> Note that we have the following equation:
> \begin{equation}
> 	\mathbb{E}[L_{P_Z}(\theta_t)]=\mathbb{E}[L_{S_{\mathrm{l}},\hat{S}_{\mathrm{u},t}}(\theta_t)]+\mathrm{gen}_t.
> \end{equation}
> On the LHS, it is the population risk (i.e., test loss) which we aim to control. On the RHS, the first term is the empirical training loss (cf. (5)) and the second term is the generalization error. Since we already consider the empirical risk minimization (ERM) based algorithms that minimize the first term on the RHS, controlling $\mathrm{gen}_t$ is equivalent to controlling the test loss. Moreover, in (5), we include \emph{both} the labeled data training loss and the pseudo-labeled data training loss. Thus,  when the training loss converges, the expected test loss will be bounded as well.
>
> **Q3**: Thanks. In this area, researchers are typically  interested in upper bounds because we want to control the generalization error in terms of input and output distributions, algorithms, loss functions and etc. As far as we know, there are only a limited number of works about the lower bounds for generalization error, whose settings and definitions are also quite different from what we study. In the papers by Kinouchi \& Caticha (1993) and Gr{\o}nlund et al. (2019), they either defined the generalization error as the error probability or the gap between the test and training error probabilities, which is different from our definitions. Other papers by Seroussi \& Zeitouni (2021) and Mahdavi et al.\ (2015) only derived lower bounds for the population risk or the excess risk under certain settings. From the information-theoretic viewpoint, there is no existing work about the lower bounds for generalization error.  Therefore, in the future there is still a lot of potential to investigate the lower bounds, but this is certainly out of the scope of the current paper as it has hitherto not been done in the completely supervised setting.
>
> **Q4**: 	In our paper, for any loss function $l:\mathcal{Z} \times \Theta \to \mathbb{R}$ and any iteration $t$, we define the generalization error as
> $\mathrm{gen}\_t=\mathbb{E}[ L_{P_Z}(\theta_t)-L_{S_{\text{l}},\hat{S}_{\text{u},t}}(\theta_t)]$
>
> $=w(\mathbb{E}\_{\theta_t}[\mathbb{E}\_{Z}[l(\theta_t,Z) | \theta_t]]-\frac{1}{n}\sum_{i=1}^n \mathbb{E}\_{\theta_t,Z_i}[l(\theta_t,Z_i)] )$
> $+ (1-w)(\mathbb{E}\_{\theta_t}[\mathbb{E}\_{Z}[l(\theta_t,Z) | \theta_t]]-\frac{1}{m}\sum_{i=(t-1)m+1}^{tm}\mathbb{E}\_{\theta_t,X'_i,\hat{Y}'_i}[l(\theta_t,(X'_i,\hat{Y}'_i))] )$
>
> which is the expected gap between the population risk and empirical risk, and takes value on $\mathbb{R}$. It is not a probability so it is not bounded by 0 and 1. In the bGMM classification problem, we consider the negative log-likelihood loss function which takes value on $\mathbb{R}$ and in Figure 6(c) we can see that the empirical generalization error ranges from 0 to 5.
>
> **[Part II continued in the following response]**

---

> > ### Author Response · Authors · 2021-11-14
> > **Response to Reviewer 5JBw [Part II]**
> >
> > **Q5**: 	The reviewer is absolutely right concerning this point. Indeed, as can be seen in Theorem 1, the upper bound is applicable to *all* algorithms characterized by the mappings $\\{P_{\theta_k | S_{\mathrm{l}},S_{\mathrm{u}}} \\}\_{k=0}^t$ and pseudo-labelling functions $\\{f_{\theta_k}\\}\_{k=0}^{t-1}$. To make sense of this bound, we particularize it to the the bGMM classification problem with iterative SSL setting in which the pseudo-labelling function is $f_{\mathbf{\theta}\_{t-1}}(\mathbf{X}\_i')=\mathrm{sgn}(\mathbf{\theta}\_{t-1}^\top \mathbf{X}\_i')$ for any unlabeled data sample $\mathbf{X}_i'$ at iteration $t$. Moreover, compared to bounds based on the VC dimension or the uniform stability  that depend on the complexity of the model family, the information-theoretic bounds concern about algorithms, data distributions and loss functions.
> >
> > We hope that the clarifications above are satisfactory and sincerely hope that the esteemed reviewer is open to upping the score.
> >
> > ----
> > **References**
> >
> > Shai Shalev-Shwartz,  Shai Ben-David. Understanding machine learning: From theory to algorithms. Cambridge university press, 2014.
> >
> > Osame Kinouchi and Nestor Caticha. Lower bounds on generalization errors for drifting rules. Journal of Physics A: Mathematical and General, 26(22): 6161, 1993.
> >
> > Allan Gr{\o}nlund,  Lior Kamma,  Kasper Green Larsen, Alexander Mathiasen  and  Jelani Nelson. Margin-based generalization lower bounds for boosted classifiers. In Advances in Neural Information Processing Systems, 32: pp 11963-11972, 2019.
> >
> > Inbar Seroussi  and Ofer Zeitouni. Lower Bounds on the Generalization Error of Nonlinear Learning Models. arXiv preprint arXiv:2103.14723, 2021.
> >
> > Mehrdad Mahdavi, Lijun Zhang and Rong Jin. Lower and upper bounds on the generalization of stochastic exponentially concave optimization. In Conference on Learning Theory, pp 1305-1320, 2015.

---

### Decision · Program_Chairs · 2022-01-20

**Decision:**

Reject

**Comment:**

This paper studies the generalization error of semi-supervised learning, where the algorithm gradually pseudo-labels the data throughout the learning process. Theoretically, an upper bound on the generalization error is shown to decompose into a term that vanishes with successive labeling and another that does not, leading to a plateau in performance. This is studied analytically for a mixture of two Gaussians. Experimentally, similar behavior is also observed to occur in more realistic scenarios. What reviewers struggled with is to understand what part of the results are, to some extent, obvious, and what offer deeper insight. What is obvious: even if a Bayes classifier were available for pseudo-labeling, feature overlap means that there is a plateau of noise beyond which labeling cannot improve. What is not obvious: is it even worth pseudo-labeling, or could we make things worse? The merit of the paper is in elucidating the latter. There are several concerns that remain, however, even after discussions. First, there is whether the insight is substantial or not. Here, some comparison and contrast with existing literature suggests otherwise. Second, there is whether the experimentally observed behavior is an instance of the phenomenon described by theory. Here, better structured experiments are needed to tie in with the theory. Overall, although the paper presents compelling insight, it is not yet ready to disseminate. It needs a stronger argument for its added theoretical contribution and clearer experiments to support that the presented theory is indeed behind the empirical behavior of these iterative algorithms.